# tRF-1:30-Gly-CCC-3 inhibits thyroid cancer via binding to PC and modulating metabolic reprogramming

Bifei Fu[1],*, YuMing Lou[1],*, Xiaofeng Lu[1], Zhaolin Wu[3], Junjie Ni[1], Cong Jin[1], Pu Wu[2],† , Chaoyang Xu[1,2],†

tRFs and tiRNAs (tRNA-derived fragments) are an emerging class of small noncoding RNAs produced by the precise shearing of tRNAs in response to specific stimuli. They have been reported to regulate the pathological processes of numerous human cancers. However, the biofunction of tRFs and tiRNAs in the development and progression of papillary thyroid cancer (PTC) has not been reported yet. In this study, we aimed to explore the biological roles of tRFs and tiRNAs in PTC and discovered that a novel 5′tRNA-derived fragment called tRF-1:30-Gly-CCC-3 (tRF-30) was markedly down-regulated in PTC tissues and cell lines. Functionally, tRF-30 inhibited the proliferation and invasion of PTC cells. Mechanistically, tRF-30 directly bound to the biotin-dependent enzyme pyruvate carboxylase (PC), downregulated its protein level, interfered with the TCA cycle intermediate anaplerosis, and thus affected metabolic reprogramming and PTC progression. These findings revealed a novel regulatory mechanism for tRFs and a potential therapeutic target for PTC.

## Introduction

Thyroid cancer (TC) is the most prevalent endocrine cancer, and its incidence rate has been rapidly increasing during the past few decades (Chen et al, 2023). Papillary thyroid cancer (PTC) comprises 80% of all malignant thyroid tumors, whereas additional histological subtypes include follicular, anaplastic, and medullary TCs (Filetti et al, 2019). Although the overall 5–10-yr survival rate of PTC has reached 80–95% with conventional thyroidectomy followed by radioiodine treatment and TSH suppression therapy, 20–30% of PTC can dedifferentiate into a more aggressive phenotype, referring to increasing tumor metastasis and resistance to therapy with only 10% of the 10-yr survival rate (Luo et al, 2021). Therefore, it is critical to identify novel molecular mechanisms and specific therapeutic targets for PTC carcinogenesis. For now, with increasing malignancies metabolically characterized, there seem to be new

therapeutic strategies that target cancer metabolism (Park et al, 2020; Stine et al, 2022). The metabolic mechanism of TC, however, is poorly described. Therefore, more studies are needed to better understand the mechanisms underlying the metabolic modification of PTC and develop new strategies for aggressive PTC.

Metabolic reprogramming is a crucial hallmark of cancers and plays a critical role in cancer development. The Warburg effect has long been considered as the primary metabolic reprogramming in cancer cells (Sun et al, 2018; Siska et al, 2020; Ohshima & Morii, 2021). For a few years, the biotin-dependent enzyme pyruvate carboxylase (PC), which catalyzes the carboxylation of pyruvate to oxaloacetate, has been linked to metabolic reprogramming in multiple cancer models by replenishment of the TCA cycle intermediates (Cappel et al, 2019; Kiesel et al, 2021; Sheng et al, 2022). Recently, further studies have revealed that PTC was characterized as the active TCA cycle (Strickaert et al, 2019). In addition, Liu et al observed that the expression of PC in tumor samples and fine-needle aspiration washout of PTC with lymph node metastasis was significantly increased (Liu et al, 2021a; Liu et al, 2021b). However, the regulatory mechanism and biological function of PC-mediated metabolic reprogramming in PTC progression remain unclear.

tRNA-derived small noncoding RNAs (tRNA fragments, tRFs, and tiRNAs) have emerged as novel tumor targets and have steadily become the focus of tumor research. tiRNAs, containing 5′-tRNA halves (5′tiRNA) and 3′-tRNA halves (3′tiRNA), were generated from the cleavage of the anticodon loop by ribonuclease. Based on where they cleave from tRNAs, tRFs are divided into four types: tRF-1, tRF-2, tRF-3, and tRF-5. A, B or C subtypes of tRF-5 and tRF-3 can be further categorized (Yu et al, 2020; Fu & Xu, 2022). Recent studies have shown that some tRFs may be involved in altering the metabolic pathways underlying breast cancer progression (Zhu et al, 2021; Liu et al, 2022a). In almost all solid tumors, researchers have discovered robust correlations between abnormally expressed tRF-related mRNAs and glucose metabolism (Mo et al, 2019; Li et al, 2021). Moreover, some pieces of evidence have shown that aberrantly expressed tRFs in PTC tissues (Han et al, 2021; Shan et al, 2021), however, deeper mechanistic insight into the association between tRFs and tumorigenesis of PTC is needed, which may open

[1]Department of Breast and Thyroid Surgery, Affiliated Jinhua Hospital, Zhejiang University School of Medicine, Jinhua, China   [2]Central Laboratory, Affiliated Jinhua Hospital, Zhejiang University School of Medicine, Jinhua, China   [3]Department of Anaesthesiology, Affiliated Jinhua Hospital, Zhejiang University School of Medicine, Jinhua, China

Correspondence: xuchaoyang@zju.edu.cn; wupucr@163.com
*Bifei Fu and YuMing Lou contributed equally to this work
†Pu Wu and Chaoyang Xu are jointly supervised this work

a new avenue to establish novel diagnostic biomarkers or promising therapeutic targets.

In this study, using the tRF- and tiRNA-sequencing technique, we aimed to explore the role and underlying mechanism of tRNA fragments in PTC by identifying their differential expression in tumor tissues compared with adjacent normal tissues. The result showed that tRF-5c was primarily down-regulated, indicating that tRF-5c may be critical in inhibiting the occurrence and progression of PTC. In the subsequent quantitative real-time polymerase chain reaction (qRT–PCR) validation, a tRF-5c type tRF-1:30-Gly-CCC-3 (labeled in the MINTbase as tRF-30-PNR8YO9LON4V) was found to be significantly down-regulated in PTC tissues. This tRF was also named tRF-30, as it is spliced from the 30th nucleotide of tRNA-Gly. Moreover, we discovered that tRF-30 suppressed the proliferation and invasion of PTC cells. For the first time, we mechanistically identified that tRF-30 directly bound to PC, reduced its protein stability, and disrupted the replenishment of the TCA cycle, thus affecting metabolic reprogramming and inhibiting PTC progression. These findings provide new metabolic pathways and potential therapeutic targets for the onset and development of PTC.

# Results

## Differential expression of tRFs and tiRNAs in PTC tissues and paired normal tissues

To determine the expression of tRFs and tiRNAs in PTC, four pairs of PTC tissues (T) and adjacent normal tissues (AT) were collected for a high-throughput sequencing experiment. The specific workflow is summarized in Fig 1A. The principal component analysis and the correlation coefficient analysis showed that there was a significantly different expression between the two groups (Fig 1B and C). A total of 140 commonly expressed tRFs and tiRNAs were detected in tumor tissues and 125 in ATs, and 110 of which were detected in both groups (Fig 1D). The component analysis revealed that tRFs were the primary down-regulated type in PTC, whereas tiRNAs were the slightly outnumbered type (Fig 1E), indicating that tRFs may play a more significant role in PTC. The volcano plot and heatmap showed that a total of 35 considerably dysregulated tRFs were detected in the tumor group. Of these, 20 were present at particularly low levels in tumor tissues compared with ATs (Fig 1F and G) (Table S1).

To further identify tRFs as prospective target molecules, the distributions of tRFs with differential expression in tumor tissues and ATs were analyzed. The result showed that tRF-5c was the main down-regulated type (Fig 1H), indicating that tRF-5c may function as a potential tumor suppressor of PTC. Considering the parameters of tRF-5c, such as $P$-value, FC-value, and length, tRF-30 was selected for further qRT–PCR verification in 30 pairs of tumor tissues and ATs. The result showed that tRF-30 was notably down-regulated in tumors (Fig 1I). Therefore, we hypothesized that tRF-30 might play an inhibiting role in PTC. Furthermore, the expression of tRF-30 was detected in the human PTC cell lines BCPAP, TPC1, KTC1, and NPA87, related to the Nthy-ori 3-1 normal thyroid epithelial cell line. The result showed that BCPAP and KTC1 were the lowest two cell lines in tRF-30 expression (Fig 1J). Moreover, we detected the knockdown

models of tRF-30 in TPC1 and NPA87 cells by qRT–PCR assay, and the results showed that the knockdown efficiencies of the cells were unacceptably low (Fig S1A and B). Therefore, BCPAP and KTC1 cell lines were chosen for further biological function research and downstream mechanism studies.

## tRF-30 inhibits proliferation and invasion in PTC cells and regulates tumor growth in a mouse model

To further explore the biological function of tRF-30, the overexpression and knockdown models were constructed in BCPAP and KTC1 cells. The transfection efficiencies were detected by qRT–PCR assay (Fig 2A and B). Cell counting kit-8 (CCK-8) assay indicated that tRF-30 overexpression significantly decreased the viability of BCPAP and KTC1 cells. In contrast, tRF-30 knockdown markedly increased cell viability (Fig 2C and D). Colony formation assay demonstrated that tRF-30 overexpression dramatically reduced colony formation in BCPAP and KTC1 cells, whereas tRF-30 knockdown considerably boosted colony formation in cells (Fig 2E and F). Transwell invasion assay revealed that tRF-30 overexpression significantly inhibited the invasion of BCPAP and KTC1 cells, whereas tRF-30 knockdown significantly promoted cell invasion (Fig 2G and H).

To further confirm these effects of tRF-30 in vivo, ($1 \times 10^7$, 200 $\mu l$) BCPAP cells stably transfected with tRF-30 or negative control were subcutaneously injected into the right upper back of the nude mice. Tumor volumes were measured every week and tumors were dissected after 5 wk (Fig 2I). Tumor volumes and weights were markedly decreased in the oe-tRF-30 group compared with the oe-NC group (Fig 2J and K). Moreover, subcutaneous tumors were further deployed for Ki-67 immunohistochemistry (IHC) assays. The result showed that the oe-tRF-30 group had considerably lower tumor proliferation (Fig 2L). These data revealed that tRF-30 transfection inhibited tumor growth of PTC in vivo. Collectively, both in vitro and in vivo assays demonstrated that tRF-30 acted as a suppressive factor in PTC.

## tRF-30 directly binds to PC and down-regulates PC protein by decreasing its stability in PTC cells

As has been widely reported, tRFs contribute to tumor progression by interacting with target RNAs or proteins (Xie et al, 2020; Yu et al, 2021), so we speculated that tRF-30 would exert a suppressive effect by associating with target proteins. RNA-pulldown assay was performed to verify the hypothesis (Fig 3A). The biotin-labeled tRF-30 and its antisense probes were synthesized and incubated with protein lysates of BCPAP cells to identify distinct protein bands. The results from two independent tRF-30 pulldown assays detected a 130-kD band on the silver staining gel (Fig 3B). Furthermore, 208 specific binding proteins were selected by mass spectrometry because of their presence in the band containing the tRF-30 sequence, but not that containing the antisense sequence (Table S2). Among them, PC was verified by three independent pulldown experiments and Western blot analysis (Fig 3C). In addition, RNA immunoprecipitation (RIP) assays also confirmed that tRF-30 was enriched when an antibody against PC was used in BCPAP and KTC1 cell lysates, compared with the IgG antibody group (Fig 3D).

Figure 1. **An overview of tRNA fragments in PTC tissues and adjacent normal tissues.**

**(A)** The workflow of tRF and tiRNA sequencing and analysis. **(B)** The principal component analysis of tRF and tiRNA expression in four pairs of PTC tissues (T) and adjacent normal tissues (AT). **(C)** The correlation coefficient analysis of tRF and tiRNA expression in tumor tissues and ATs. **(D)** Venn diagram of tRFs and tiRNAs detected only in tumor tissues (red region), only in ATs (blue region), and in both tissues (purple region). **(E)** Distributions of different tRFs and tiRNAs in tumor tissues and ATs. **(F)** Heatmap of differentially expressed tRFs and tiRNAs in the two groups. **(G)** Volcano plot of down-regulated tRFs and tiRNAs (green dots) and up-regulated tRFs and tiRNAs (red dots) in tumor tissues compared with ATs. Fold change ≥ 1.5 and *P*-value < 0.05. **(H)** Differentially expressed tRFs and tiRNAs were separately calculated in PTC,

The above mentioned studies have proved that tRF-30 could specifically interact with PC, but more research into the interaction's molecular processes is required. The qRT–PCR assay results showed that tRF-30 overexpression and knockdown had no significant effect on the PC mRNA expression in BCPAP and KTC1 cells (Fig 3E). Moreover, the Western blot assay indicated that tRF-30 adversely regulated PC protein levels in cells (Fig 3F). Therefore, we speculated that tRF-30 might regulate PC protein stability in PTC cells. Cycloheximide (CHX, 50 µg/ml) was used to stop protein (including PC) synthesis in BCPAP and KTC1 cells. The results showed that tRF-30 transfection enhanced the degradation of PC protein, whereas sh-tRF-30 transfection prolonged the half-life of PC protein in cells. GAPDH was referred as a control (Fig 3G and H). The protein degradation mediated by the ubiquitin–proteasome pathway is the primary mechanism for the regulation of intracellular protein levels, which contributes to the degradation of more than 80% of intracellular protein (Wang & Maldonado, 2006). The results of the in vitro ubiquitination assay in Fig S2 displayed that tRF-30 modulated the degradation of PC protein in BCPAP and KTC1 cells via the ubiquitination pathway. Collectively, these above mentioned outcomes demonstrated that tRF-30 directly bound to PC, regulated ubiquitin/proteasome-dependent degradation, and decreased the stability of PC protein in PTC cells.

### PC promotes the proliferation and invasion of PTC cells and is up-regulated in PTC tissues

It has been confirmed in previous research that PC expression is dysregulated in several tumor types, including lung cancer, breast cancer, PTC (Wang et al, 2021; Liu et al, 2022b; Chen et al, 2022), etc. As shown in Fig 4A, PC was significantly elevated in PTC tissues. Then, we confirmed that expression levels of PC mRNA (Fig 4B) and protein (Fig 4C) were much higher in BCPAP cells than in KTC1 and TPC1 cells, and lowest in Nthy-ori 3-1 cells, suggesting the potential pro-tumorigenic role for PC in PTC. We therefore selected BCPAP and KTC1 cells to investigate the carcinogenic effect of PC. The overexpression and knockdown efficiencies were detected by qRT–PCR and Western blot (Fig 4D and E). The viability of BCPAP and KTC1 cells was significantly higher than that of vector mock-transfected control cells after PC overexpression, as shown by the CCK-8 assay. In contrast, PC knockdown markedly reduced cell viability (Fig 4F and G). In addition, PC overexpression significantly increased colony formation in BCPAP and KTC1 cells, whereas PC knockdown greatly reduced colony formation in cells (Fig 4H and I). Transwell invasion assay showed that the invasion ability of PC-overexpressed BCPAP and KTC1 cells was considerably increased compared with the control groups, whereas PC knockdown markedly reduced the invasion ability of cells (Fig 4J and K). These results suggested that PC could be an oncogenic driver in PTC, which was inconsistent with the impact of tRF-30 on PTC.

### PC participates in the regulation of tRF-30 on PTC

We then confirmed PC expression levels in PTC tissues compared with ATs by IHC assay. The results showed that the PTC group had obviously higher PC levels than the AT group (Fig 5A), which was consistent with the previous studies (Fig 4A). To further validate whether tRF-30 regulates the biological behaviors of PTC via its association with PC, we performed a "rescue" experiment by transfecting tRF-30 with or without PC overexpression in BCPAP and KTC1 cells. After combined transfection, cell proliferation and invasion were detected, and CCK-8 assay results showed that PC transfection enhanced cell viability. Relative to the oe-tRF-30 group, cell viability in the oe-tRF-30 + oe-PC group was increased, which presented that PC transfection weakened the influence of tRF-30 overexpression on cell viability (Fig 5B). Similar results were found for the cell invasion experiment. Fig 5C and D illustrated that PC transfection elevated cell invasion, and alleviated the oe-tRF-30–resulted reduction of the invasion ability. Moreover, in vivo assays showed that PC expression in the oe-tRF-30 group was significantly decreased compared with that in the oe-NC group (Fig 5E). In summary, we speculated that the effects of tRF-30 on the proliferation and invasion of PTC cells were dependent on PC, the expression of which was regulated by tRF-30.

### tRF-30 mediates PC to decrease TCA cycle metabolites and regulate metabolic reprogramming in PTC cells

PC is a biotin-dependent enzyme, which is important for replenishing TCA cycle intermediates and mediating metabolic reprogramming in eukaryotes (Valle, 2017). It would be intriguing to investigate whether tRF-30 affects the TCA cycle and mediates metabolic reprogramming when it binds to PC in PTC cells. To further verify the metabolic regulation of tRF-30, a metabolomic analysis using liquid chromatography–tandem mass spectrometry (LC–MS/MS) was performed in BCPAP cells with or without tRF-30 transfection. The specific workflow is summarized in Fig 6A. The principal component analysis results indicated that there was an ideal significantly different expression between the two groups (Fig 6B). A total of 34 commonly expressed metabolites were clustered by sample groups and metabolin classes (Fig 6D). Notably, the volcano plot and heatmap revealed that tRF-30 considerably decreased the levels of citrate, oxaloacetate (OAA), the primary intermediates of the TCA cycle, and ornithine, the product of TCA-related urea cycle, in BCPAP cells. In addition, tRF-30 remarkably increased the levels of pyruvate, the main substrate of PC carboxylation, and guanosine, the precursor of TCA-related GTP synthesis, in BCPAP cells (Fig 6C and E). These modulated metabolites were involved in metabolic signaling pathways, including the TCA cycle, pyruvate metabolism, amino acid metabolism, and 2-oxocarboxylic acid metabolism (Fig 6F). Dysfunction of PC blocks the transition of pyruvate to OAA, which was consistent with our results that overexpression of tRF-30 resulted in the decrease of

---

and tRF-5c was the main down-regulated type (13/20). **(I)** Relative expression of tRF-30 in 30 pairs of tumor tissues and ATs was detected by qRT–PCR. **(J)** Relative expression of tRF-30 was detected by qRT–PCR in normal thyroid epithelial cells (Nthy-ori 3-1) and four PTC cell lines (BCPAP, TPC1, KTC1, and NPA87). Data were shown as mean ± SD. (*t* test and Mann–Whitney *U* test, *$P < 0.05$, **$P < 0.01$, and ***$P < 0.001$). Quantitative results were based on three independent experiments. Source data are available for this figure.

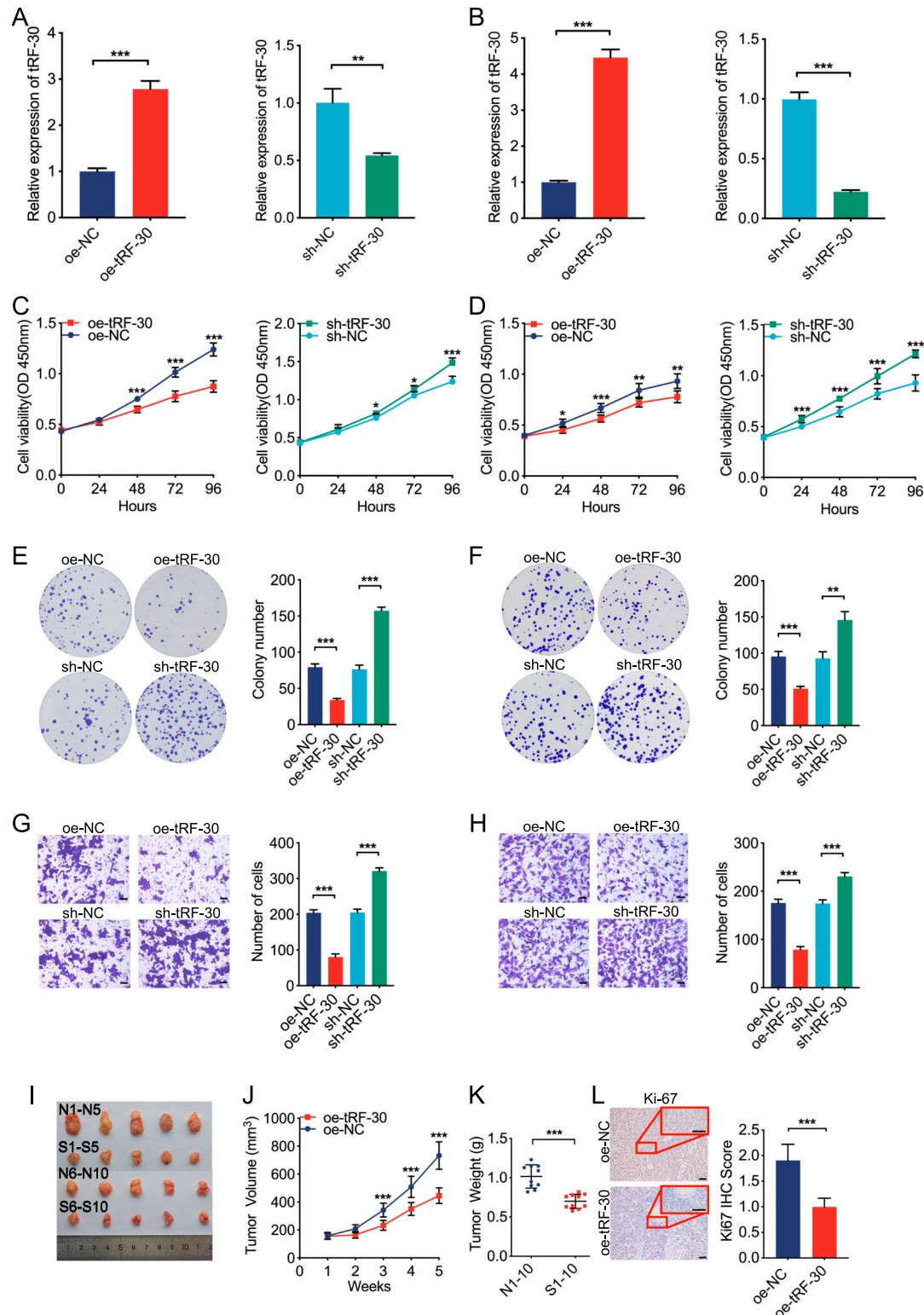

**Figure 2. tRF-30 inhibits proliferation and invasion of PTC cells in vitro and in vivo.**
**(A, B)** Overexpression and knockdown efficiencies of tRF-30 in BCPAP (A) and KTC1 (B) cells were verified by qRT–PCR. **(C, D)** tRF-30 suppressed the viability of BCPAP (C) and KTC1 (D) cells by CCK-8 assay. **(E, F)** tRF-30 inhibited the proliferation of BCPAP (E) and KTC1 (F) cells by colony formation assay. **(G, H)** tRF-30 repressed the invasion of BCPAP (G) and KTC1 (H) cells by transwell invasion assay. Scale bar = 50 μm. **(I, J, K)** Tumor growth in the oe-tRF-30 group (S1–S10) was suppressed compared with the oe-NC

OAA and other TCA cycle-associated metabolites (Fig 6G). These results suggested that tRF-30 combination with PC could inhibit the replenishment of TCA cycle intermediates and change the metabolic reprogramming in PTC cells.

To confirm whether tRF-30 regulates TCA cycle intermediates via its association with PC, we transfected tRF-30 with or without PC overexpression in BCPAP and KTC1 cells. After tRF-30 and/or PC transfection, cell citrate and pyruvate levels were detected by ELISA assays. The results showed that exogenous expression of PC could partially weaken the decreased levels of citrate and the increased levels of pyruvate in tRF-30 transfection cells (Fig 7A and B). These results revealed that inhibition of PC by tRF-30 could interfere with TCA cycle intermediate anaplerosis and dysregulated metabolic intermediates to suppress the development of PTC via altering metabolic reprogramming (Fig 7C).

Carcinoma cells show preferential use of lactate-generating glycolysis over the common route of oxidative phosphorylation. This altered metabolism, named the "Warburg effect," has been considered for a long time to be the major metabolic reprogramming in cancer. The changes in glycolytic pathways have been associated directly or indirectly with the downstream targets of many different ncRNAs (Mirzaei & Hamblin, 2020). Therefore, we further explored whether tRF regulated the Warburg effect. The glucose uptake and lactate production assay results showed that tRF-30 overexpression and knockdown had no significant effect on the glycolysis in BCPAP and KTC1 cells (Fig S3A and B). Therefore, we speculated that tRF-30 had no significant effect on the Warburg effect in PTC cells.

## Discussion

TC has become a striking health issue over the past several decades because of its increasing incidence. PTC accounts for more than 80% of all thyroid malignancies and has an excellent prognosis after standard treatments, including thyroidectomy, radioactive iodine-131, and TSH suppressive therapy (Megwalu & Moon, 2022; Houten et al, 2023). However, up to 20% and 10% of those with PTC will present with local recurrence and distant metastasis, respectively. Among these patients, two-thirds exhibit initial or gradual loss of the iodine uptake, indicating a status of dedifferentiation with limited effective treatment, which is known as being lethal for its less than 10% 10-yr survival rate (Ma et al, 2019; Ulisse et al, 2021; Liu et al, 2023). Clinically, precise diagnostic targets and efficient therapeutic methods for these patients are still insufficient. At present, profiting from the development of high-throughput sequencing technology, tRNA fragments (tRFs and tiRNAs), which are known to be formed by the specific endonuclease cutting in the selective site of tRNAs, are known to have important implications in the biological processes of numerous human cancers and may serve as potential diagnostic and therapeutic targets for regulating cancer pathological processes (Wang

et al, 2022b; Pekarsky et al, 2023). However, the mechanisms of tRFs and tiRNAs in the occurrence and development of PTC are unknown, so we performed the tRF and tiRNA sequencing in PTC.

Sequencing data from this study revealed that tRF-5c was the predominant down-regulated tRNA fragment in PTC, indicating that tRF-5c may be involved in preventing PTC progression. In contrast, tiRNAs were the slightly up-regulated type, suggesting that they may have an oncogenic function in PTC. It is the first time to acquire direct evidence of the contrary effects of various tRNA fragment types on PTC, and such conclusions may not be accidental. Besides, we identified a down-regulated tRF-5c type fragment, tRF-30 in PTC, for the first time, and its biological functions and precise mechanisms were systematically explored in the present research.

The impact of tRFs on different cancers varies widely. A tRF has been reported to promote the progression of colorectal and lung cancer (Yang et al, 2022b; Lu et al, 2022). Several tRFs were reported to be tumor suppressors in breast cancer (Goodarzi et al, 2015; Wang et al, 2022a). To the best of our knowledge, this is the first investigation into the PTC tissues' functional tRFs. tRF-30, a 30 nt tRNA fragment generated from the 5' end of the mature tRNA, remarkably decreased expression in PTC tissues and dramatically inhibited PTC cell progression in vitro and tumorigenicity in vivo. Collectively, these results reveal that tRF-30 may be a novel potential suppressor in PTC, and targeting tRF-30 may be a promising novel therapy for PTC.

Although the study of tRF function is still in its infancy, the in-depth mechanisms of tRFs in various cancers have been emerging in recent years. A recent large-scale analysis of tRF–mRNA associations in the TCGA cohort's pan-cancer types has shown various cancer-specific pathways that are universally regulated by tRFs across many cancer types (Telonis et al, 2019). With RNA-pulldown assay and RIP assay, a completely different model was discovered in the current study, in which the tRF-30 directly binds to PC and decreases its stability. PC is a biotin-dependent enzyme that catalyzes the carboxylation of pyruvate to oxaloacetate, aiming to replenish the TCA cycle, and its activity has been shown to promote metabolic reprogramming in several malignant tumor progressions (Yang et al, 2022a; Sheng et al, 2022). Our current data proved that PC was highly expressed in PTC tissues and promoted the progression of PTC cells in vitro, which functions oppositely from tRF-30. Interestingly, we observed that the interaction with tRF-30 caused a dramatic decline in PC protein, whereas having no discernible effect on PC mRNA level, suggesting that tRF-30 may destabilize PC protein in PTC cells. As expected, IHC assay results identified that PTC tissues had higher tRF-30 expression compared with the adjacent normal tissues. Data from in vivo experiments further demonstrated that PC expression in the oe-tRF-30 group was considerably lower compared with the control group. In addition, with the "rescue" experiment by transfecting tRF-30 with or without PC overexpression, we demonstrated that PC significantly attenuated the impact of tRF-30 on the proliferation and invasion of PTC cells. Notably, our study enriches the understanding of molecular

---

group (N1–N10). **(L)** tRF-30 down-regulated Ki-67 staining of PTC tumors in an in vivo mouse model by IHC assay. Scale bar = 50 μm. Data were triple replicated and shown as mean ± SD. (*t* test and Mann–Whitney *U* test, *$P < 0.05$, **$P < 0.01$, and ***$P < 0.001$). NC means negative control, sh means short hairpin RNA, and oe means overexpression. Source data are available for this figure.

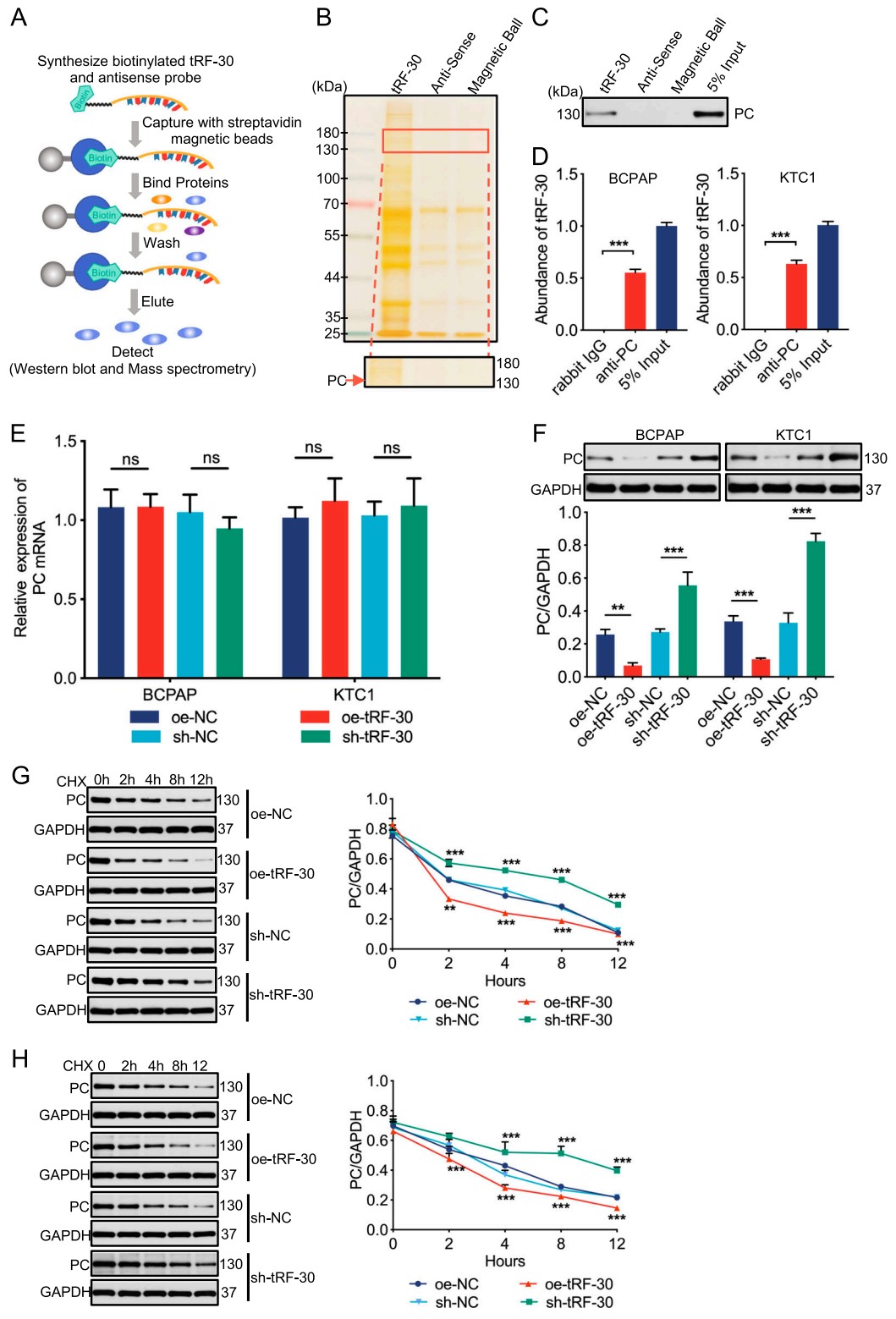

**Figure 3. tRF-30 combines with PC and regulates the stability of PC protein.**
**(A)** The workflow of the tRF-30 pulldown assay. **(B)** Differential proteins pulled down in BCPAP cells were detected by a silver staining assay. The red arrow marked the band where PC was located in the tRF-30 sense group. Anti-sense and magnetic ball were used as controls. **(C)** PC in RNA pulldown assays was verified by Western blot. 5% input acted as a positive control. **(D)** The binding between endogenous PC and tRF-30 was confirmed in BCPAP and KTC1 cells by RIP assays. Rabbit IgG acted as a negative control, and 5% input acted as a positive control. **(E, F)** After oe-tRF-30 and sh-tRF-30 transfection, the mRNA (E) and protein (F) levels of PC were tested by qRT–PCR and Western blot. GAPDH was used as a control. **(G, H)** After tRF-30 (or sh-tRF-30) transfection and CHX treatment, the PC protein level was tested in BCPAP (G) and KTC1

mechanisms by which tRFs in cancer cells bind protein partners and affect posttranscriptional regulation.

The LINC00092 has been reported to interact with PC to modulate glycolysis and oxidative stress of breast cancer cells, a novel biological mechanism by which noncoding RNA association with PC regulates cancer metabolism (Chen et al, 2022). To elucidate whether tRF-30 can mediate the metabolic mechanism of PTC via its interaction with PC, we performed LC-MS-based on metabolite levels in BCPAP cells. Notably, tRF-30 overexpression significantly decreased the levels of certain TCA cycle-related metabolites, including citrate, OAA, and ornithine. In addition, tRF-30 remarkably increased the levels of some TCA cycle precursors as pyruvate and guanosine. These metabolite analysis results indicated that the replenishment of TCA cycle intermediates was prevented by tRF-30 transfection. tRF-30 can thereby mediate the metabolic process of PTC cells. Through further in vitro metabolite assessment, we proved that the metabolic reprogramming regulated by tRF-30 was PC-dependent. Increasing evidence suggests that metabolic reprogramming via replenishment of the TCA cycle intermediates is essential for proliferating cancer cells to synthesize nucleic acids, lipids, and amino acids (Cappel et al, 2019; Kiesel et al, 2021). Therefore, these findings revealed that the metabolic reprogramming regulated by tRF-30 via its binding with PC resulted in the suppression of biological behavior in PTC cells and indicated that the level of tRF-30 might be used as a potential therapeutic target for PTC.

However, there are several limitations to this study. First, the four PTC tissues and their adjacent normal tissues used for sequencing were from a homogenous population at the Zhejiang Provincial Hospital, the sequencing data might not cover all dysregulated tRNA fragments. Moreover, more patients should be enrolled in the research to enlarge the sample size and allow for a more thorough correlation of clinical–pathological factors with tRF-30, which may uncover its prognostic value in PTC. Second, further studies are needed to investigate if tRF-30 could affect other metabolic pathways by binding other mRNAs and proteins.

In conclusion, our study revealed that tRF-30 was significantly down-regulated in PTC and it could suppress tumor progression. Mechanistically, we identified that tRF-30 directly bound to PC, mediated its protein stability, and decreased the level of PC, thus inhibiting the replenishment of the TCA cycle intermediates in PTC cells, changing metabolic reprogramming, and suppressing PTC progression. Our findings provided a new model for the regulation of tRNA fragments on PTC and a promising potential therapeutic target for PTC.

# Materials and Methods

## Patient tissue samples

A total of 50 pairs of human PTC tissues and adjacent normal tissues were collected from patients who underwent thyroidectomy in the Department of Breast and Thyroid Surgery at the Jinhua Municipal Central Hospital Medical Group between 2020 and 2022 and preserved in liquid nitrogen. All samples were pathologically diagnosed with PTC, and no patients received preoperative treatment. This study was approved by the Ethical Review Committee of Jinhua Municipal Central Hospital Medical Group and all specimens were collected with written informed consent obtained from patients.

## Human PTC cell lines

The human PTC cell lines (BCPAP, TPC1, KTC1, NPA87) and the normal thyroid epithelial cells (Nthy-ori 3-1) were purchased from the iCell Bioscience Inc. All cell lines were identified by STR analysis and removed from mycoplasma contamination. Cells were cultured in RPMI 1640 medium (Sigma-Aldrich) supplemented with 10% FBS (Gibco) and 1% penicillin–streptomycin (Gibco) at 37°C under 5% $CO_2$.

## Tissue tRF and tiRNA sequencing and analysis

To identify tRFs and tiRNAs in human PTC tissues, total RNAs were extracted from four pairs of PTC tissue and paired normal tissues using TRIpure Total RNA Extraction Reagent (ELK Biotechnology). The small RNA-sequencing libraries were constructed as the method mentioned previously (Mo et al, 2019). Briefly, total RNA samples were pretreated by 3′-aminoacyl (charged) deacylation, 5′-OH (hydroxyl group) phosphorylation, $m^1A$ demethylation, and $m^3C$ demethylation. Next, the pretreated total RNA samples were used for library construction and small RNA-sequencing. For tRF and tiRNA sequencing, 15–40 nt RNA biotypes were sequenced with the NextSeq 500/550 V2 system (#FC-404-2005; Illumina) according to the manufacturer's protocol. The tRFs and tiRNAs sequencing analysis were performed by the Arraystar tRF and tiRNA-seq data package. The above sequencing data are accessible through accession number GSE197438 in the NCBI Gene Expression Omnibus (GEO) database.

## qRT–PCR

The total RNAs of human PTC tissues and PTC cell lines were extracted using TRIpure Total RNA Extraction Reagent (ELK Biotechnology). The Bulge-loop miRNA qRT–PCR Starter Kit (RiboBio Co) was used for tRF and tiRNA reverse transcription in a 10 μl reaction volume. Then, the cDNA was amplified by qRT–PCR using EnTurbo SYBR Green PCR SuperMix (ELK Biotechnology) with the QuantStudio 6 Flex Real-Time PCR System (Life Technologies). *Actin* was used as the endogenous control for mRNA, and *U6* was used as the endogenous control for tRFs and tiRNAs. The expression of genes was calculated using the $2^{-\Delta\Delta Ct}$ method. The primer sequences were listed in Table S3.

---

(H) cells by Western blot at 0, 2, 4, 8, and 12 h. GAPDH was used as a control. Data were triple-replicated and shown as mean ± SD. (*t* test and Mann–Whitney *U* test, n.s. $P \geq 0.05$, *$P < 0.05$, **$P < 0.01$, and ***$P < 0.001$). NC means negative control, sh means short hairpin RNA, and oe means overexpression. Protein expression was quantified by ImageJ analysis of Western blots. Full blots were provided in the Original Source Data file.
Source data are available for this figure.

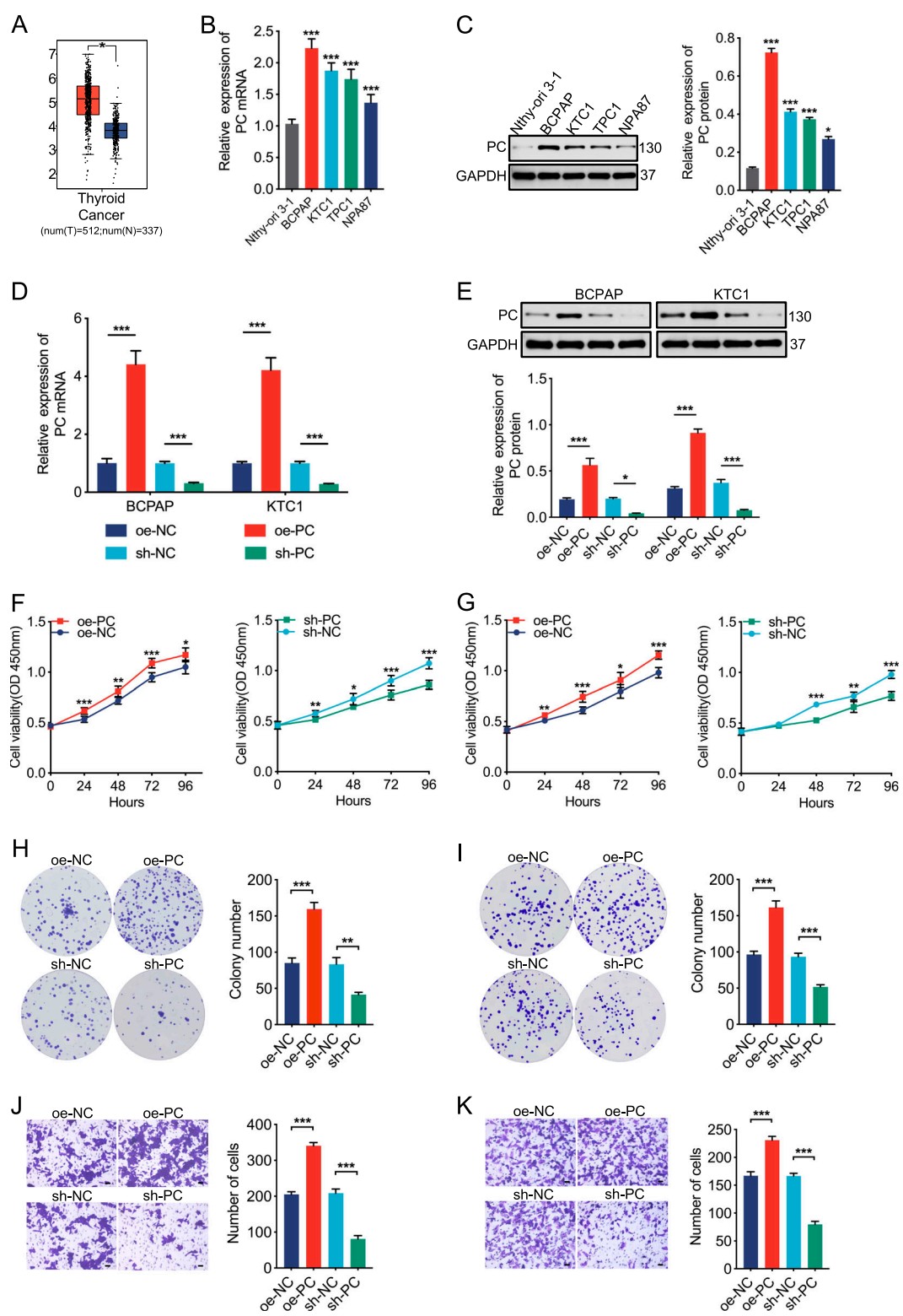

**Figure 4. PC promotes the proliferation and invasion of PTC cells in vitro.**
**(A)** Box plot showed PC was up-regulated in thyroid cancer tissues according to the database. **(B)** Relative expression of PC mRNA was detected by qRT–PCR in Nthy-ori 3-1 and four PTC cell lines. **(C)** Relative expression of PC protein was detected in Nthy-ori 3-1 and four PTC cell lines by Western blot with its quantification result. GAPDH was used as a control. **(D, E)** Overexpression and knockdown efficiencies of PC in BCPAP and KTC1 cells were verified by qRT–PCR (D) and Western blot (E). **(F, G)** PC promoted the viability of BCPAP (F) and KTC1 (G) cells by CCK-8 assay. **(H, I)** PC accelerated the colony formation of BCPAP (H) and KTC1 (I) cells by clonogenic assay. **(J, K)** PC increased the invasion abilities of BCPAP (J) and KTC1 (K) cells by transwell invasion assay. Scale bar = 50 μm. Data were triple-replicated and shown as mean ± SD. (t test

## Lentivirus preparation and cell transfection

The tRF-30 overexpression model (oe-tRF-30) and the PC over-expression model (oe-PC) were constructed by transfection with the mimic sequence. The tRF-30 knockdown model (sh-tRF-30) and the PC knockdown model (sh-PC) were constructed by transfection with the shRNA lentivirus. All the lentivirus models were designed and synthesized by QIAGEN. The PTC cells were infected with lentiviruses by using Lipofectamine 3000 reagent (Invitrogen) according to the manufacturer's protocol. All the mimic and shRNA sequences are listed in Table S4.

## CCK-8 assay

The CCK-8 assay (C0038; Beyotime) was used to detect cell pro-liferation. Briefly, cells were plated in the 96-well plates at a density of 1,000 cells per well, at fixed intervals of 0, 24, 48, 72, and 96 h; 10 $\mu$l of CCK-8 reagent was added to each well and incubated for 2 h under 37°C. The OD values at 450 nm were measured by the microplate reader (Thermo Fisher Scientific). Each assay was repeated three times.

## Colony formation assay

The plate cloning assay was used to detect cell colony formation. Briefly, Cells were seeded in the six-well plates at a density of 600–800 cells per well and cultured for seven consecutive days at 37°C under 5% $CO_2$. Then, the cells of each well were collected, fixed in 4% PFA for 30 min, stained with crystal violet for 30 min, and washed with PBS. Finally, the clones of each well were photographed under the light microscope (Olympus) and counted by the ImageJ software (NIH, Bethesda, Maryland, USA). Each assay was repeated three times.

## Transwell invasion assay

The invasion assay was carried out using the transwell chambers (Corning) with Matrigel mix (3 mg/ml) (Becton Dickinson and Co.[BD]) coated inserts. Cells were starved on day 1 by culture in the FBS-free medium. On day 2, 3.0 × 10$^5$ cells were resuspended in 500 $\mu$l of serum-free medium and inoculated into the upper chamber, and 700 $\mu$l of RPMI 1640 medium containing 20% FBS without cells was added to the lower chamber as the chemo-attractant. The transwell chambers were incubated at 37°C for 48 h under 5% $CO_2$. Then, the cells on the surface of the lower chamber were fixed with 4% PFA for 30 min, stained with crystal violet for 30 min, and washed with PBS. Finally, the chambers were photographed under the light microscope and the cells were counted by the ImageJ software. Each assay was repeated three times.

## RNA-pulldown assay and mass spectrometry analysis

Biotin-labeled tRF-30 and antisense probes were synthesized by GenScript. The Pierce Magnetic RNA-Protein Pull-Down Kit (20164; Thermo Fisher Scientific) was used to perform the RNA-pulldown assay according to the manufacturer's protocol. The bead–RNA–protein complexes were diluted in SDS buffer, developed with a Fast Silver Stain Kit (P0017S; Beyotime), and the retrieved proteins were sent for mass spectrometry analysis.

## RIP assay

The RIP assay was carried out using the RIP Kit (17-701; Millipore). Briefly, rabbit anti-PC antibody (PA5-50101; Thermo Fisher Scientific) or rabbit anti-IgG antibody (PP64B; Millipore) was integrated by the magnetic beads and incubated with the PTC cells lysate. Finally, the protein-binding RNA was retrieved and subjected to qRT–PCR analysis.

## In vitro ubiquitination assay

Briefly, after different transfections, BCPAP and KTC1 cells were treated with 10 $\mu$M MG132 for 5 h (S1748; Beyotime) to inhibit degradation of ubiquitinated proteins. RIPA lysis and extraction buffer (Solarbio) were used to split cells. The lysate supernatant was collected for co-immunoprecipitation with anti-PC antibody (PA5-50101; Thermo Fisher Scientific) or anti-IgG antibody and Protein A/G agarose (Thermo Fisher Scientific) with gentle rocking at 4°C overnight. After centrifugation and precipitation, the su-pernatant was subjected to Western blotting. Antibodies against the following were used for Western blotting: mouse anti-PC an-tibody (sc-271493; Santa Cruz Biotechnology), mouse anti-ubiquitin antibody (3936; CST). Rabbit anti-GAPDH antibody (5174; CST) was used as the endogenous control.

## Western blot analysis

Total cell protein was extracted using RIPA lysis and extraction buffer (Solarbio) supplemented with a protease inhibitor cocktail (P8340; Roche). After quantification of the protein concentration by the BCA protein detection kit (Beyotime), 20 $\mu$g protein lysates were loaded and separated with 10% SDS–PAGE (Epizyme), and then the protein bands were transferred onto a PVDF membrane (Millipore). The membrane was blocked with 5% skim milk for 1 h, and then incubated at 4°C overnight with mouse anti-PC antibody (sc-271493; Santa Cruz Biotechnology). Rabbit anti-GAPDH antibody (5174; CST) was used as the endogenous control. Then, the membrane was washed with Tris-buffered saline supplemented with Tween 20 (TBST) three times and incubated with the corresponding sec-ondary antibody (Proteintech) for 2 h under room temperature and washed three times with TBST. Finally, the membrane was visu-alized with ECL reaction reagents (AS1059; Aspen) and detected by

and Mann–Whitney $U$ test, *$P$ < 0.05, **$P$ < 0.01, and ***$P$ < 0.001). Protein expression was quantified by ImageJ analysis of Western blots. Full blots were provided in the Original Source Data file.
Source data are available for this figure.

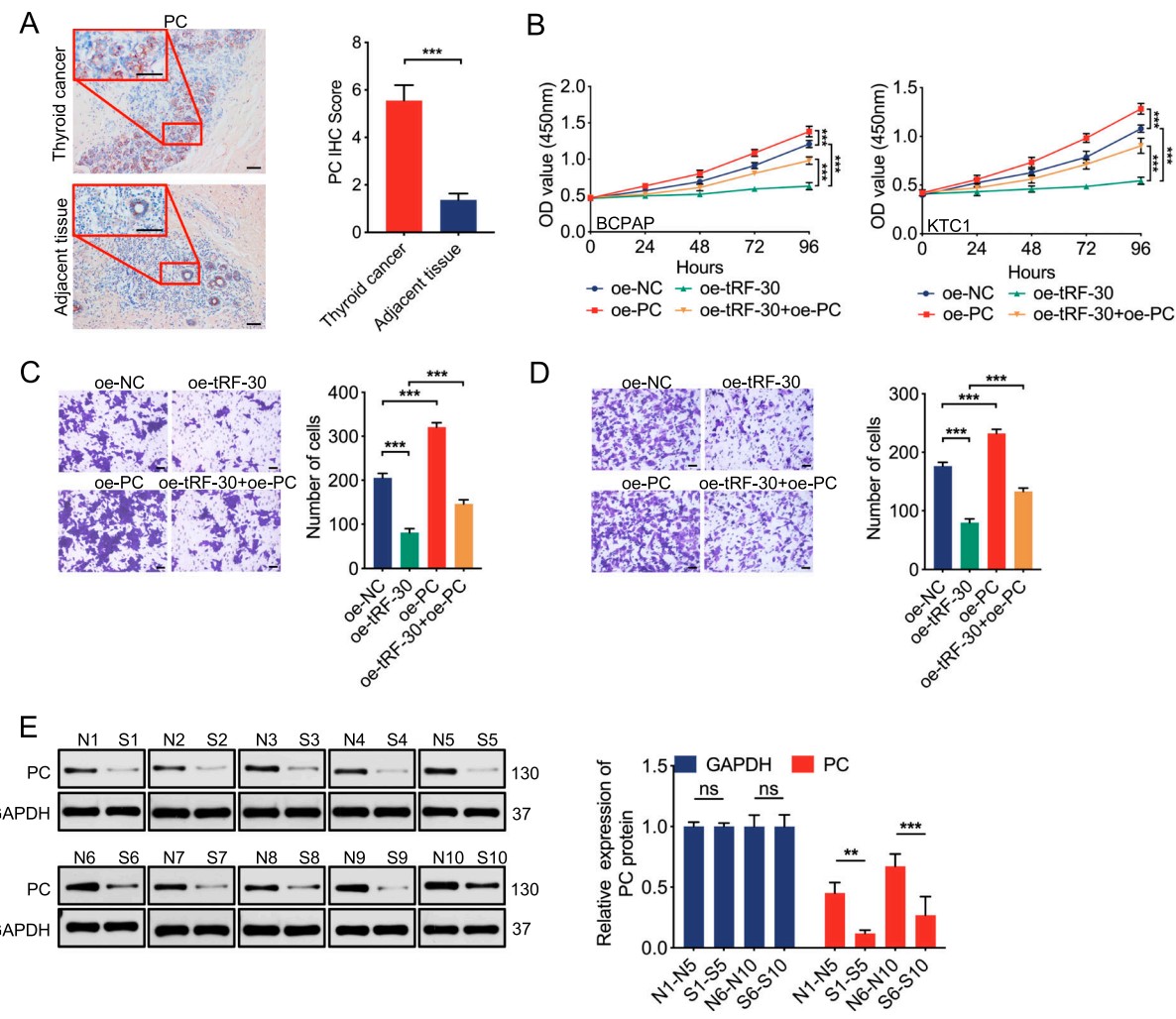

**Figure 5. The relationship between tRF-30 and PC is proven in vitro and in vivo.**
**(A)** PC expression was up-regulated in thyroid cancer tissues by an IHC assay. Scale bar = 50 $\mu$m. **(B)** CCK-8 assay results showed that the effect of tRF-30 on cell viability was partly compromised by PC overexpression in BCPAP and KTC1 cells. **(C, D)** Rescue experiments showed that the tRF-30 effect on transwell invasion assay was partially attenuated by PC overexpression in BCPAP (C) and KTC1 (D) cells. Scale bar = 50 $\mu$m. **(E)** PC expression of mice subcutaneous tumors was down-regulated in the oe-tRF-30 group (S1–S10) compared with the oe-NC group (N1–N10) by Western blot and its quantification results. GAPDH was used as a control. Data were triple-replicated and shown as mean ± SD. ($t$ test and Mann–Whitney $U$ test, n.s. $P \geq 0.05$, *$P < 0.05$, **$P < 0.01$, and ***$P < 0.001$). NC means negative control, sh means short hairpin RNA, and oe means overexpression. Protein expression was quantified by ImageJ analysis of Western blots. Full blots were provided in the Original Source Data file. Source data are available for this figure.

the Western Lightning Gel Imaging System (GE). Quantitative analysis of the individual band was measured by ImageJ software.

### IHC staining

The IHC staining assay was carried out using the IHC Kit (Zsgb Bio) according to the manufacturer's protocol. Briefly, the PFA-fixed and paraffin-embedded tissues were sectioned. The sections were deparaffinized in xylene, hydrated in ethanol with descending concentrations, blocked the endogenous peroxidase activity with serum, and incubated with primary antibodies at 4°C overnight and secondary antibodies for 1 h under room temperature. Finally, the sections were stained by DAB reagent and H&E staining (DAB substrate chromogen system; Dako); the slides were sealed and images were obtained with the microscope. Antibodies were used

against the following: rabbit anti-Ki67 antibody (NBP2-22112; Novus) and mouse anti-PC antibody (sc-271493; Santa Cruz Biotechnology).

### In vivo tumorigenesis model

The 5-wk-old female BALB/c nude mice were chosen for in vivo tumorigenesis to study the effect of tRF-30 on tumor growth. The nude mice were obtained from Vital River Laboratories and fed in the Experimental Animal Center of Jinhua Municipal Central Hospital Medical Group. The nude mice were randomly divided into each group (N = 10 mice per group). $1 \times 10^{7}$ BCPAP cells stably transfected with tRF-30 or control sequence were subcutaneously injected into the right upper back skin area of the nude mice. The tumor volumes were measured every week. After 5 wk, the tumors were dissected for weight measurement, IHC staining, and WB analysis. All animal

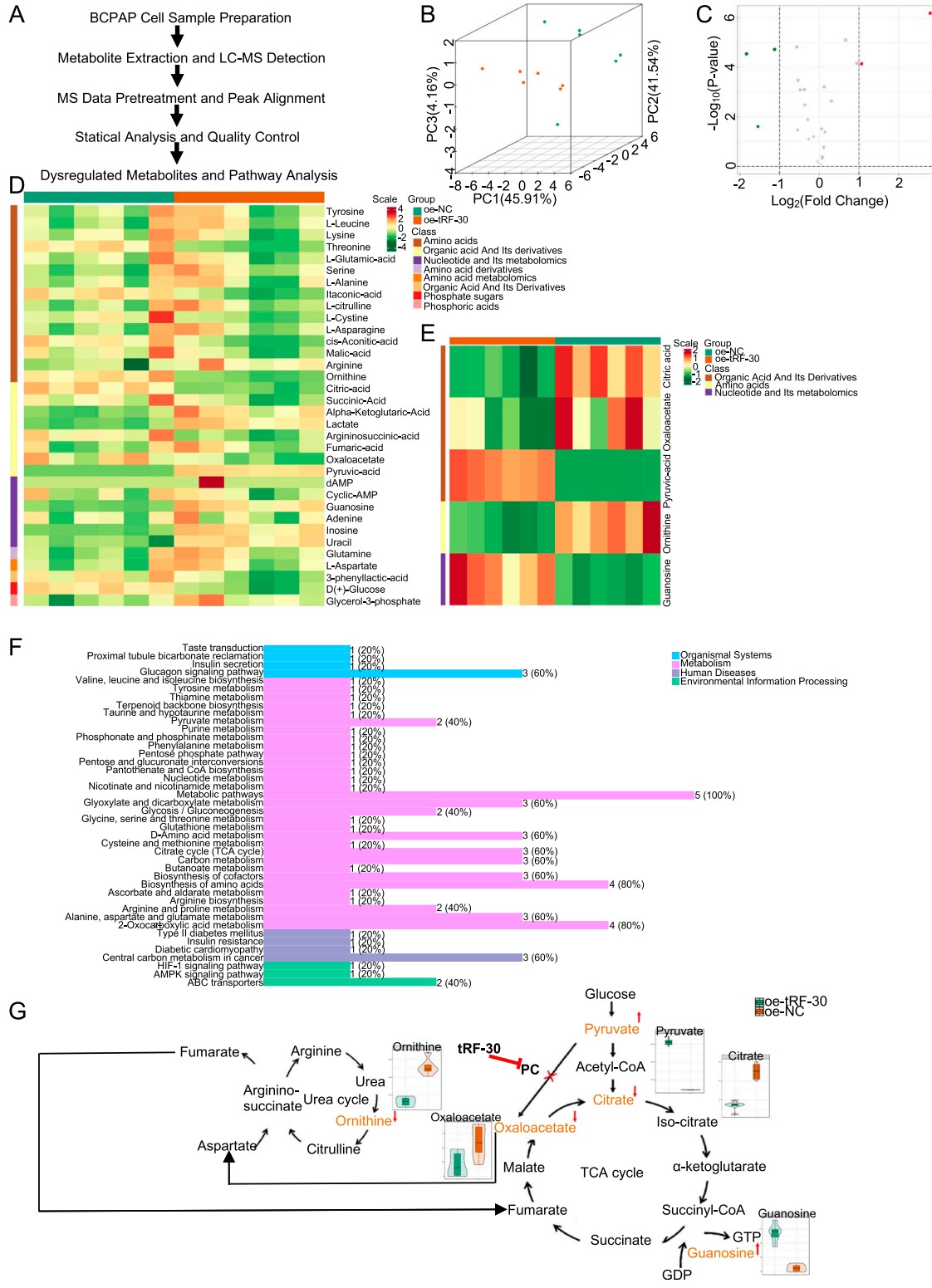

**Figure 6. Metabolomic analysis of the effects of tRF-30 on BCPAP cells.**
**(A)** The workflow of LC-MS-based metabolomic analysis. **(B)** The principal component analysis of metabolite expression in the oe-tRF-30 group compared with the oe-NC group. N = 6 per group. **(C)** Volcano plot of differentially expressed metabolite induced by tRF-30. Red dots represent significantly up-regulated metabolites, and green dots represent significantly down-regulated metabolites in the oe-tRF-30 group. Fold change ≥ 2 and VIP > 1. **(D)** Heatmap clustering displayed hierarchical metabolites in the two groups. **(E)** Heatmap hierarchical clustering annotated major differential metabolites in the oe-tRF-30 group compared with the oe-NC group. Fold change ≥ 2 and *P* < 0.05. **(F)** KEGG pathway enrichment analysis of differential metabolites involved in metabolic processes and signaling pathways between the two groups. **(G)** Summary of tRF-30-induced changes in metabolic pathways based on the results of metabolome analysis.
Source data are available for this figure.

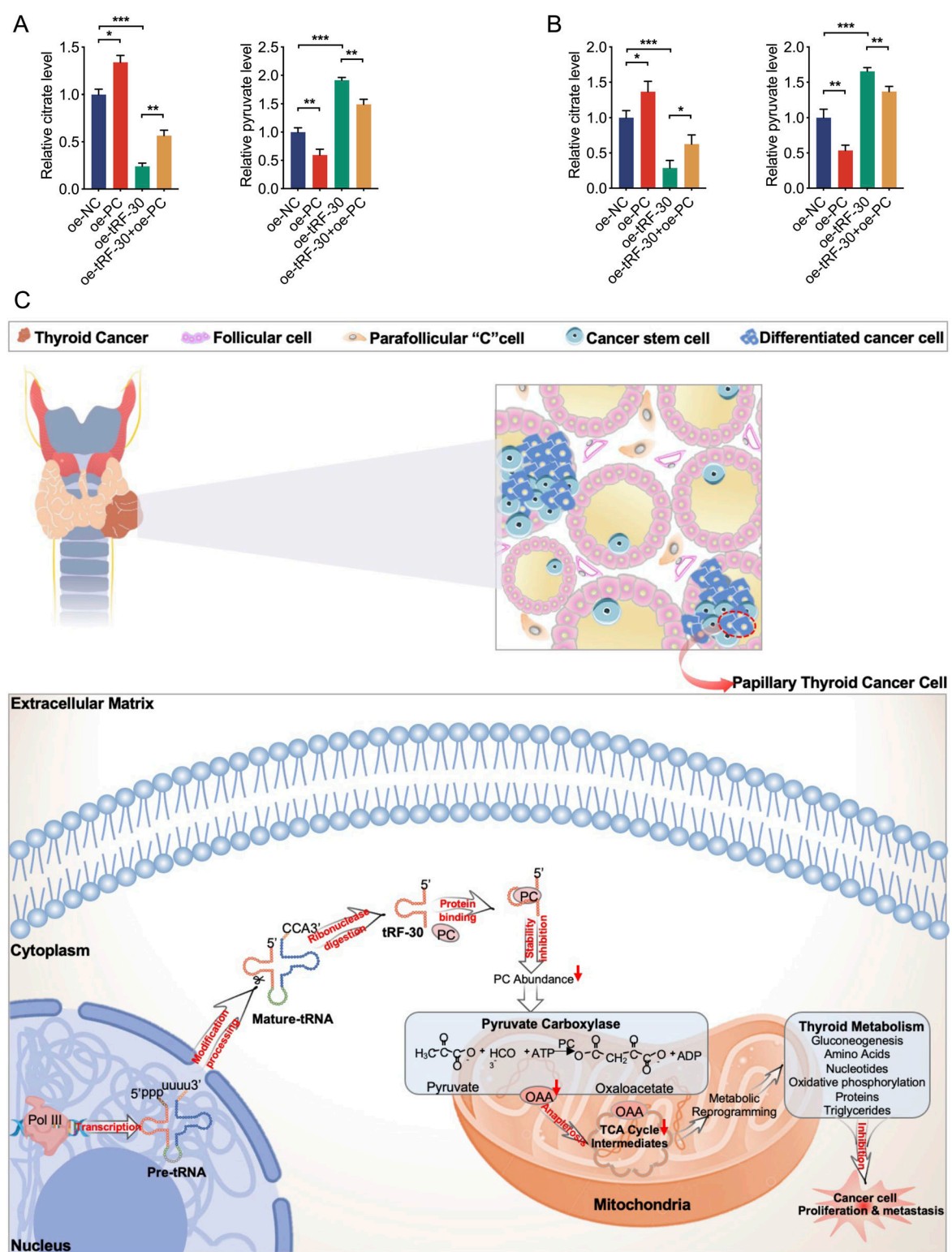

**Figure 7. Overexpression of PC weakened the influence of tRF-30 transfection on the metabolic intermediates.**
**(A, B)** ELISA assay results showed that the effect of tRF-30 on citrate and pyruvate levels was partially attenuated by PC overexpression in BCPAP (A) and KTC1 (B) cells.
**(C)** The schematic diagram of the mechanisms by which tRF-30 directly bound to PC, decreased its protein stability, and interfered with replenishment of the TCA cycle intermediates, thus affecting metabolic reprogramming and suppressing PTC progression. Data were triple-replicated and shown as mean ± SD. (*t* test and Mann–Whitney *U* test, *$P < 0.05$, **$P < 0.01$, and ***$P < 0.001$). NC means negative control, and oe means overexpression.

experiments were approved by the Ethical Review Committee of Jinhua Municipal Central Hospital Medical Group.

### Cell pyruvate assay

The pyruvate assay kit (ab65342; Abcam) was used to measure pyruvate concentration in cells according to the manufacturer's protocol. Briefly, $5 \times 10^6$ BCPAP cells or KTC1 cells were homogenized in the pyruvate assay buffer and de-proteinized using a 10-kD spin filter (Amicon Ultra-0.5 ml Centrifugal Filter) with centrifugation at 14,000$g$ for 10 min under 4°C. Then, the filtrate was added by the reaction mix and the plate was incubated at RT for 30 min under the dark. Finally, the mixture was measured at 587 nm with a microplate reader (model 680; Bio-Rad) and the concentration of pyruvate was interpolated from the standard curve. Treatment groups were calculated relative to the control group with a value of 1.

### Cell citrate assay

The citrate assay kit (ab83396; Abcam) was used to assess citrate levels in cell lysates according to the manufacturer's protocol. Briefly, $5 \times 10^6$ BCPAP cells or KTC1 cells were resuspended in 100 $\mu$l of assay buffer and centrifuged at 16,000$g$ for 5 min under 4°C. Then, the supernatant was de-proteinized using a 10 kD spin filter (Amicon Ultra-0.5 ml Centrifugal Filter) with centrifugation at 14,000$g$ for 10 min under 4°C. Then, the filtrate was added by the reaction mix and the plate was incubated at RT for 30 min under the dark. Finally, absorbance at 570 nm was detected by a microplate reader (model 680; Bio-Rad) and the concentration of citrate was interpolated from the standard curve. Treatment groups were calculated relative to the control group with a value of 1.

### Lactate production and glucose uptake measurement

$3.0 \times 10^5$ cells were inoculated onto six-well plates and incubated at 37°C for 24 h. Afterwards, 5 $\mu$l of the cell culture supernatant were collected in 96-well plates and mixed with 200 $\mu$l of the Glucose Assay Reagent (Sigma-Aldrich) or 100 $\mu$l of Lactate Assay Regent (Sigma-Aldrich). After incubation at 37°C for 20 min, the mixture was read calorimetric at the specific wavelength and the concentration of glucose and lactate was interpolated from the standard curve.

### Targeted metabolomic analysis

The cell metabolites were extracted by the manufacturer's protocol. In brief, adding precooled methanol extraction solvent (Merk), vortexing for 3 min, freeze–thawing with liquid nitrogen, and an ice water bath for 10 min. The cell suspension was centrifuged at 12,000$g$ for 10 min under 4°C and the supernatant was filtered through the 3 kD cutoff filter (Amicon Ultra, Merk KGaA); finally, the filtrate was centrifugally concentrated to LC–MS/MS analysis. The metabolites of the TCA cycle, glycolysis, and other intracellular metabolites were determined using LC–MS on a tandem mass spectrometer coupled to ultra-performance liquid chromatography (Ex-ion LC AD). Sample quality was guaranteed by multiple reaction monitoring.

### Statistical analysis

All the results were presented as mean ± SE. Statistical analysis was carried out using SPSS 26.0 (IBM). $t$ test, Mann–Whitney $U$ test, and analysis of variance (ANOVA) were performed to compare the statistical significance between two or more groups. A paired $t$ test was performed to compare the expression of tRF-30 in 30 PTC tissues and paired normal tissues. For all data, a $P$-value < 0.05 was statistically significant (*$P$ < 0.05, **$P$ < 0.01, and ***$P$ < 0.001).

## Data Availability

The data discussed in this article have been deposited in NCBI's Gene Expression Omnibus and are accessible through GEO Series accession number GSE197438. Each sample is GSM5917412, GSM5917413, GSM5917414, GSM5917415, GSM5917416, GSM5917417, GSM5917418, and GSM5917419.

### Ethics statement

All patients gave informed consent, and the study was performed in accordance with the Committee for Ethical Review of Research of Jinhua Municipal Central Hospital Medical Group. All animal experiments were approved by the Committee for Ethics of Animal Experiments of Jinhua Municipal Central Hospital Medical Group.

## Supplementary Information

## Acknowledgements

This work was supported by Project of Zhejiang Provincial Medical Health Science and Technology (Grant No.: 2023KY1284), Jinhua Science and Technology Research Program (Grant No.: 2022-3-073), Jinhua Public Welfare Technology Application Project Research (Grant No.: 2023-4-076), and Basic Research Foundation project of Jinhua Central Hospital (Grant No.: JY2022-2-15).

### Author Contributions

B Fu: conceptualization, data curation, formal analysis, investigation, visualization, methodology, and writing—original draft, review, and editing.
Y Lou: formal analysis, funding acquisition, investigation, visualization, methodology, and writing—review and editing.
X Lu: conceptualization, formal analysis, supervision, investigation, and writing—review and editing.
Z Wu: data curation, software, formal analysis, visualization, methodology, and writing—review and editing.
J Ni: formal analysis, investigation, methodology, and writing—review and editing.
C Jin: formal analysis, supervision, investigation, and writing—review and editing.

P Wu: conceptualization, supervision, funding acquisition, investigation, project administration, and writing—original draft, review, and editing.

C Xu: conceptualization, resources, supervision, funding acquisition, project administration, and writing—original draft, review, and editing.

## Conflict of Interest Statement

The authors declare that they have no conflict of interest.

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
