## [Reviewer comments · Life Science Alliance]

Life Science Alliance

tRF-1:30-Gly-CCC-3 inhibits thyroid cancer via binding to PC and modulating metabolic reprogramming

Bifei Fu, Yuming Lou, Xiaofeng Lu, Zhaolin Wu, Junjie Ni, Cong Jin, Pu Wu and Chaoyang Xu

DOI: <https://doi.org/10.26508/lsa.202302285>

Corresponding author(s): Chaoyang Xu (Affiliated Jinhua Hospital, Zhejiang University School of Medicine, Jinhua)

Review Timeline:

Submission Date:	2023-07-19
Editorial Decision:	2023-09-26
Revision Received:	2023-11-04
Editorial Decision:	2023-12-01
Revision Received:	2023-12-05
Accepted:	2023-12-05

Transaction Report:

September 26, 2023

Re: Life Science Alliance manuscript #LSA-2023-02285-T

Prof. Chaoyang Xu
Affiliated Jinhua Hospital, Zhejiang University School of Medicine, Jinhua
CHINA

Dear Dr. Xu,

Thank you for submitting your manuscript entitled "tRF-1:30-Gly-CCC-3 inhibits thyroid cancer via binding to PC and modulating metabolic reprogramming" to Life Science Alliance. The manuscript was assessed by an expert reviewer, whose comments are appended to this letter. We invite you to submit a revised manuscript addressing the Reviewer comments.

When submitting the revision, please include a letter addressing the reviewer comments point by point.

Thank you for this interesting contribution to Life Science Alliance. We are looking forward to receiving your revised manuscript.

Sincerely,

B. MANUSCRIPT ORGANIZATION AND FORMATTING:

Reviewer #1 (Comments to the Authors (Required)):

This article is an interesting discovery. Through high-throughput sequencing, the author selected tRF-1: 30-Gly-CCC-3 with low expression in thyroid cancer for research and verified that this tRF can inhibit the malignant progression of thyroid cancer through experiments such as cck8 and transwell. The tRF-binding protein PC was found by Pull down assay, and it was found that the tRF could down-regulate the expression of the protein. Finally, the authors found that the tRF can regulate the metabolic reprogramming of PTC by interfering with the TCA cycle. In summary, the author confirmed the role of tRF in PTC and found novel regulatory mechanisms. I recommend accepting the publication of this article, but there are still some small problems that need to be improved before the final publication.

1. The authors found that tRF-5c was the main down-regulated type, but did not clearly explain how to directly select tRF-1: 30-Gly-CCC-3 for subsequent verification based on factors such as P-value, FC-value, and length.
2. The author selected the two cells with the lowest expression for subsequent experiments, but generally, we will choose the cells with relatively high expression for a knockdown, please explain the reason why TPC1 or NPA87 was not selected for a knockdown.
3. The too-long naming of the tRF in Figure affects the cleanliness and beauty, so it is recommended to simplify its naming.
4. In Figure 2I, whether the S and N groups are marked reversely.
5. Please check the format of the symbols in the text.
6. How does this tRF specifically regulate its stability by binding to PC protein?
7. The Warburg effect is an important metabolic feature of tumor cells. The author found that the tRF can reduce the level of several important intermediates in the TCA cycle, and whether the tRF will have an effect on the Warburg effect.

Dear Editor,

Thank you very much for providing us with the comments on our manuscript entitled “tRF-1:30-Gly-CCC-3 inhibits thyroid cancer via binding to PC and modulating metabolic reprogramming”. We thank the reviewer for the time spent reviewing our manuscript and for the highly constructive and helpful comments that enabled us to further improve the quality of our manuscript. We did our utmost to address all issues that were raised, as listed point-by-point below.

Reviewer #1 (Comments to the Authors (Required)):

This article is an interesting discovery. Through high-throughput sequencing, the author selected tRF-30 with low expression in thyroid cancer for research and verified that this tRF can inhibit the malignant progression of thyroid cancer through experiments such as cck8 and transwell. The tRF-binding protein PC was found by Pull down assay, and it was found that the tRF could down-regulate the expression of the protein. Finally, the authors found that the tRF can regulate the metabolic reprogramming of PTC by interfering with the TCA cycle. In summary, the author confirmed the role of tRF in PTC and found novel regulatory mechanisms. I recommend accepting the publication of this article, but there are still some small problems that need to be improved before the final publication.

1. The authors found that tRF-5c was the main down-regulated type but did not clearly explain how to directly select tRF-1:30-Gly-CCC-3 for subsequent verification based on factors such as P-value, FC-value, and length.

Response: Dear reviewer, thanks for this precious suggestion. As suggested, we have provided raw data on significantly dysregulated tRNA fragments in the Supplementary Materials section in **RED** as follows:

A form of tRF-5c known as tRF-1:30-Gly-CCC-3 has a length longer than 18 bp, a high FC-value of 0.202, and a low p-value of 0.005. Additionally, this tRF has now

been included in the Mintbase. After a comprehensive analysis of the aforementioned elements, tRF-1:30-Gly-CCC-3 was selected for subsequent verification.

tRF_ID	tRF_Seq	Annotation				log2FC	DE Statistic				
		Type	tRFdb_ID	MINTbase_ID	Length		AlignInfo	Fold_Change	p_value	q_value	Ca_CPM
IRF-1:35-Ala-AGC-4-M7	GGGGAGTGTAGCTCAG	IRF-5a	-	-	15	IRNA-Ala-AGC-4-1,GGCGATGT	-5.091825764	0.029282331	0.000165126	0.018659191	0.000273175
IRF-41:62-chrM_Ser-GCT	TAAACAACATGGCTTCTCACCA	IRF-3b	-	IRF-22-UF04Q2	22	chrM:IRNA12-SerGCT,GAGAAA	-4.682935419	0.038910337	0.008560004	0.119150845	-1.923232318
IRF-1:15-chrM_Ser-TGA	GAAAAGTCATGGAG	IRF-5a	-	-	15	chrM:IRNA17-SerTGA,GAAAAA	-3.915443236	0.06627262	0.00080732	0.119150845	-0.441145576
IRF-14:27-chrM_Ser-GCT	AGAACTCCTAACTC	IRF-2	-	-	14	chrM:IRNA12-SerGCT,GAGAAA	-3.842551873	0.069707038	0.018607357	0.161740876	-1.212199224
IRF-1:30-Gly-CCC-2	CCGCCCTGGTGTAGTGGTATCATGCAAGA	IRF-5c	5007c	IRF-30-Q1Q89P	30	IRNA-Gly-CCC-2-1,GGCCGCTC	-3.76552269	0.073530026	4.28717E-05	0.009688998	4.170830943
IRF-1:15-Gly-CCC-1-M4	GCATTGGTGGTTCAG	IRF-5a	-	-	15	IRNA-Gly-CCC-1-1,GCATTGGT	-3.44267718	0.091970999	0.008962674	0.119150845	0.30856733
IRF-1:29-Gly-CCC-2	GCGCCCTGGTGTAGTGGTATCATGCAAG	IRF-5c	-	IRF-29-Q1Q89P	29	IRNA-Gly-CCC-2-1,GGCCGCTC	-3.25471122	0.104769361	0.00068045	0.03844545	2.489183768
IRF-1:16-chrM_Ser-TGA	GAAAAGTCATGGAGG	IRF-5a	-	IRF-16-OH1690	16	chrM:IRNA17-SerTGA,GAAAAA	-3.20288979	0.108601069	0.024268847	0.168465424	0.580230884
IRF-1:31-Val-CAC-3	GTTCCCTGAGTGTAGCGGTTATCACATCCG	IRF-5c	-	IRF-31-79MP9P	31	IRNA-Val-CAC-3-1,GTTCCCTG	-3.018001369	0.12344999	0.002541661	0.095735909	2.559174777
IRF-1:15-Glu-CTC-1-M4	TCCCTCGTGGTCTAG	IRF-5a	-	-	15	IRNA-Glu-CTC-1-1,TCCCTGGT	-2.960374554	0.128480868	0.0206513	0.166685494	0.209909397
IRF-1:31-Val-AAC-2	GTTCCCTGAGTGTAGTGGTATCACGTTCCG	IRF-5c	5026c	IRF-31-79MP9P	31	IRNA-Val-AAC-2-1,GTTCCCTG	-2.440172258	0.184261011	0.027748854	0.168465424	1.03244443
IRF-1:30-Gly-CCC-1-M4	GCATTGGTGGTTCAGTGGTAGAATTCTCGC	IRF-5c	-	IRF-30-PN8Y0	30	IRNA-Gly-CCC-1-1,GCATTGGT	-2.319347696	0.200358039	0.001631507	0.073744129	14.8335607
IRF-1:30-Gly-CCC-3	GCATTGGTGGTTCAGTGGTAGAATTCTCGC	IRF-5c	-	IRF-30-PN8Y0	30	IRNA-Gly-CCC-3-1,GCATTGGT	-2.309898994	0.201675873	0.004826265	0.119150845	4.073819156
IRF-1:32-Val-CAC-3	GTTCCCTGAGTGTAGCGGTTATCACATCCG	IRF-5c	-	IRF-32-79MP9P	32	IRNA-Val-CAC-3-1,GTTCCCTG	-2.230687486	0.213057171	0.010841445	0.128956134	5.332791207
IRF-1:30-Gly-GCC-4	GCATAGTGGTTCAGTGGTAGAATTCTTCG	IRF-5c	-	-	30	IRNA-Gly-GCC-4-1,GCATAGGT	-2.207571541	0.216498429	0.006846265	0.119150845	4.878456006
IRF-1:32-Val-AAC-2	GTTCCCTGAGTGTAGTGGTATCACGTTCCG	IRF-5c	-	IRF-32-79MP9P	32	IRNA-Val-AAC-2-1,GTTCCCTG	-2.205996175	0.216734966	0.010780388	0.128956134	6.34201617
IRF-1:30-Gly-GCC-1	GCATAGTGGTTCAGTGGTAGAATTCTCGC	IRF-5c	-	IRF-30-PN8Y0	30	IRNA-Gly-GCC-1-1,GCATAGGT	-1.82221724	0.282786031	0.016087931	0.151494663	11.41350606
IRF-1:31-Glu-TTC-1	TCCCATATGGTACGGGTTAGGATTCCTGG	IRF-5a	-	IRF-31-86V8WP	31	IRNA-Glu-TTC-1-1,TCCCATAT	-1.688842206	0.310175748	0.023391606	0.168465424	8.002167241
IRF-1:30-Glu-CTC-1-M2	TCCCTGGTGTCTAGTGGTATGAGATTCGGC	IRF-5c	-	IRF-30-87R8WP	30	IRNA-Glu-CTC-1-1,TCCCTGGT	-1.679460423	0.312199318	0.026763983	0.168465424	10.5052464
IRF-1:31-His-GTG-1	CCGCTGATGTATAGTGGTATGACTCTCGG	IRF-5c	-	IRF-31-PW58VP	31	IRNA-His-GTG-1-1,CCGCTGAT	-1.625200233	0.324164894	0.031213682	0.17459312	9.116608171

Table S3: Data of significantly dysregulated tRNA fragments in PTC

2. The author selected the two cells with the lowest expression for subsequent experiments, but generally, we will choose the cells with relatively high expression for a knockdown, please explain the reason why TPC1 or NPA87 was not selected for a knockdown.

Response: Dear reviewer, thank you for the precious suggestion. The expression of tRF-30 was significantly lower in human PTC tissues than in normal tissues, so we selected the two PTC cell lines with the lowest expression of tRF-30, which was similar to the expression pattern of tRF-30 in PTC tissues. Moreover, the knockdown efficiencies of tRF-30 in TPC1 or NPA87 cells were far from satisfactory. The results have been added to the Results section in RED as follows.

We detected the knockdown models of tRF-30 in TPC1 and NPA87 cells by qRT-PCR assay, and the results showed that the knockdown efficiencies of the cells were unacceptably low (Fig S1). Therefore, BCPAP and KTC1 cell lines were chosen for further biological function research and downstream mechanism studies.

Figure S1:

Figure S1: The knockdown models of tRF-30 were not applicable in NPA87 and TPC1 cells.

(A, B) knockdown efficiencies of tRF-30 in NPA87 (A) and TPC1 (B) cells were verified by qRT-PCR. Data were triple replicated and shown as mean \pm SD. (Student's *t* test, n.s. $P \geq 0.05$). NC means negative control, sh means short hairpin RNA

3. The too-long naming of the tRF in Figure affects the cleanliness and beauty, so it is recommended to simplify its naming.

Response: Dear reviewer, thank you for the precious suggestion. As suggested, we have simplified the long naming of the tRF-1:30-Gly-CCC-3 as tRF-30 in the revised manuscript.

4. In Figure 2I, whether the S and N groups are marked reversely.

Response: Dear reviewer, thank you for the precious suggestion, and we apologize for our oversight. As suggested, we have rewritten the group names in this revised Figure 2I.

5. Please check the format of the symbols in the text.

Response: Dear reviewer, thank you very much for your constructive comments to strengthen this manuscript. As suggested, we have corrected the format of the symbol in the revised manuscript.

6. How does this tRF specifically regulate its stability by binding to PC protein?

Response: Dear reviewer, thank you very much for your specific comments. Based on our studies, tRF-30 combines with PC and down-regulates the stability of PC protein. We then further performed the ubiquitination assay to determine the mechanism of PC destabilization. The results have been added in the Results section in RED as follows:

Following treatment with CHX (the protein synthesis inhibitor), tRF-30 transfection promoted the proteasome-dependent degradation of PC in PTC cells, which decreased the stability of PC protein. Given that protein degradation mediated by the ubiquitin proteasome pathway is the primary mechanism for the regulation of intracellular protein levels, we further conducted an *in vitro* ubiquitination assay, and the results of the *in vitro* ubiquitination assay in Figure S2 displayed that tRF-30 modulated the degradation of PC protein in BCPAP and KTC1 cells via the ubiquitination pathway. Collectively, these above outcomes demonstrated that tRF-30 directly bound to PC, regulated ubiquitin/proteasome-dependent degradation, and decreased the stability of PC protein in PTC cells.

Figure S2

Figure S2: tRF-30 modulated degradation of PC protein via the ubiquitination pathway.

(A) *In vitro* ubiquitination assay was carried out to detect the ubiquitination level of PC in BCPAP and KTC1 cells. NC means negative control, sh means short hairpin RNA, oe means overexpression.

7. The Warburg effect is an important metabolic feature of tumor cells. The author found that the tRF can reduce the level of several important intermediates in the TCA cycle, and whether the tRF will have an effect on the Warburg effect.

Response: Dear reviewer, thank you for the precious suggestion. As suggested, we further explored whether tRF regulated the Warburg effect. An increase in lactate production and glucose uptake are the two common indexes reflecting the Warburg effect. Therefore, we conducted glucose uptake and lactate production assays, and the results have been added in the Results section in **RED** as follows:

Carcinoma cells show preferential use of lactate-generating glycolysis over the common route of oxidative phosphorylation (OXPHOS). This altered metabolism, named the “Warburg effect”, has been considered for a long time to be the major metabolic reprogramming in cancer. The changes in glycolytic pathways have been associated directly or indirectly with the downstream targets of many different ncRNAs (Mirzaei H & Hamblin MR, 2020). Therefore, we further explored whether tRF regulated the Warburg effect. The glucose uptake and lactate production assay results showed that tRF-30 overexpression and knockdown had no significant effect on the glycolysis in BCPAP and KTC1 cells (Fig S3, n.s. $P > 0.05$). Therefore, we speculated that tRF-30 had no significant effect on the Warburg effect in PTC cells.

Figure S3:

Figure S3: tRF-30 had no significant effect on the Warburg effect. (A) Extracellular lactate production in BCPAP and KTC1 cells. **(B)** Glucose uptake was measured in BCPAP and KTC1 cells. Data were triple replicated and shown as mean \pm SD. (Student's t test and Mann-Whitney U test, n.s. $P \geq 0.05$). NC means negative control, sh means short hairpin RNA, and oe means overexpression.

Thank you again for your evaluation.

Sincerely yours,

Chaoyang Xu, PhD. M.D

Department of Breast and Thyroid Surgery, Affiliated Jinhua Hospital, Zhejiang
University School of Medicine, Jinhua 321000, Zhejiang Province, China.

Tel.: (+86-579) 89133995; Fax: (+86-579) 89133995

E-mail: xuchaoyang@zju.edu.cn

ORCID: 0000-0001-7223-5288

December 1, 2023

RE: Life Science Alliance Manuscript #LSA-2023-02285-TR

Prof. Chaoyang Xu
Affiliated Jinhua Hospital, Zhejiang University School of Medicine, Jinhua
Department of Breast and Thyroid Surgery, Affiliated Jinhua Hospital, Zhejiang University School of Medicine
Jinhua 321000
China

Dear Dr. Xu,

Thank you for submitting your revised manuscript entitled "tRF-1:30-Gly-CCC-3 inhibits thyroid cancer via binding to PC and modulating metabolic reprogramming". We would be happy to publish your paper in Life Science Alliance pending final revisions necessary to meet our formatting guidelines.

- please upload your main manuscript text as an editable doc file
- please add ORCID ID for the secondary corresponding author -- you should have received instructions on how to do so
- please add the Twitter handle of your host institute/organization as well as your own or/and one of the authors in our system
- there is only one panel in Figure S2, so there is no need to label it as panel A. Please correct the figure, its legend, and call-out in the manuscript text
- please add callouts for Figures S1A, B and S3A, B to your main manuscript text
- the contribution of Chaoyang Xu and co-authors Yuming Lou and Pu Wu, is listed as activity, which does not qualify them for authorship. Please either update the contributions in our system and in the Author Contributions section of the manuscript, or let us know if any authors should be removed from the authorship listing.

Figure Checks:

- please add sizes next to all blots

A. FINAL FILES:

B. MANUSCRIPT ORGANIZATION AND FORMATTING:

Sincerely,

Reviewer #1 (Comments to the Authors (Required)):

The author answered all the questions I asked in great detail and revised the article accordingly, I believe that the article has met the criteria for publication and recommend acceptance of the article.

December 5, 2023

RE: Life Science Alliance Manuscript #LSA-2023-02285-TRR

Prof. Chaoyang Xu
Affiliated Jinhua Hospital, Zhejiang University School of Medicine, Jinhua
Department of Breast and Thyroid Surgery, Affiliated Jinhua Hospital, Zhejiang University School of Medicine, Jinhua 321000,
Zhejiang Province, China
Jinhua 321000
China

Dear Dr. Xu,

Thank you for submitting your Research Article entitled "tRF-1:30-Gly-CCC-3 inhibits thyroid cancer via binding to PC and modulating metabolic reprogramming". It is a pleasure to let you know that your manuscript is now accepted for publication in Life Science Alliance. Congratulations on this interesting work.

DISTRIBUTION OF MATERIALS:

Again, congratulations on a very nice paper. I hope you found the review process to be constructive and are pleased with how the manuscript was handled editorially. We look forward to future exciting submissions from your lab.

Sincerely,
